# Factors Affecting Weight Reduction after Intragastric Balloon Insertion: A Retrospective Study

**DOI:** 10.3390/healthcare11040600

**Published:** 2023-02-17

**Authors:** Mohammed A. Bawahab, Khaled S. Abbas, Walid M. Abd El Maksoud, Reem S Abdelgadir, Khaled Altumairi, Awadh R. Alqahtani, Hassan A. Alzahrani, Muneer Jan Bhat

**Affiliations:** 1Surgery Department, Faculty of Medicine, King Khalid University, P.O. Box 641, Abha 61421, Saudi Arabia; 2General Surgery, Abha International Private Hospital, Abha 62521, Saudi Arabia; 3Abha International Private Hospital, Abha 62521, Saudi Arabia; 4Surgery Department, Faculty of Medicine, King Saud University, Riyadh 11461, Saudi Arabia; 5Anesthesia, Surgery Department, Faculty of Medicine, King Khalid University, Abha 61421, Saudi Arabia

**Keywords:** intragastric balloon, obesity, excess weight loss

## Abstract

Background and Objectives: Intragastric balloon (IGB) is a safe option for obesity management. However, studies determining the factors influencing the procedure’s outcomes are scarce. Therefore, our goal was to determine the factors affecting weight reduction after IGB insertion. Materials and Methods: This retrospective study included 126 obese patients who underwent IGB treatment using the ORBERA^®^ Intragastric Balloon System. Patients’ records were retrieved; and demographic data, initial body mass index (BMI), complications, compliance with both diet and exercise programs, and percentage of excess weight reduction were recorded. Results: The study included 108 female (85.7%) and 18 male (14.3%) patients. The mean age was 31.7 ± 8.1 years. The percentage of excess weight loss (EWL) was 55.8 ± 35.7%. The mean weight loss was 13.01 ± 7.51 kg. A significant association was found between EWL and age, initial weight, initial body mass index, and the number of pregnancies. No major complications were observed. However, the balloon had to be removed early in two patients (1.59%) due to its rupture and in two other patients (1.59%) due to severe gastritis. Conclusions: IGB therapy is a safe and effective option for obesity management, associated with low rates of complications. The EWL after IGB insertion is significantly higher among older patients, those with a relatively low initial body mass index, those with a longer duration of IGB insertion, and female patients with less parity. Larger prospective studies are needed to support our results.

## 1. Introduction

Obesity is a very common growing health problem all over the world and may be considered a pandemic [1,2]. The World Health Organization (WHO) reports that more than 1.9 billion adults aged 18 years and older are overweight. Of these, over 600 million are obese. In addition, approximately 15% of women worldwide are considered obese [3]. This increased incidence of obesity is frequently associated with obesity-related comorbidities, including type 2 diabetes, hypertension, coronary heart disease, stroke, gallbladder disease, and certain types of cancer [4].

The recommended lines of treatment for obesity include lifestyle modifications, pharmacological therapy, and bariatric surgery [5]. The lack of effective medical treatments for obesity has made interventional weight loss therapies the main line of treatment against this pandemic [5,6]. Surgical intervention is recommended for patients with a body mass index (BMI) of 40 kg/m^2^, or even lower if they have failed conservative treatment or have serious comorbid conditions [7,8].

Intragastric balloon (IGB) treatment is considered a safe option for class I obesity. It is also used as a bridging procedure for patients who are morbidly obese before a bariatric surgery [9]. It is a nonsurgical, short-term modality for weight loss, which shows acceptable efficacy in weight reduction [10]. IGB induces satiety by reducing the gastric capacity and slowing the gastric emptying. Stretching of the gastric wall may also be responsible for a vagally controlled reflex that causes satiety [10,11]. However, several side effects have been reported, ranging from simple reactions including nausea and vomiting, to more serious adverse effects, such as pancreatitis, gastric perforation, and bowel obstruction [9,10,12,13].

Although some studies reported that there is a significant and clinically relevant improvement in weight loss and health outcomes associated with the use of IGB [10,14,15], a recent meta-analysis [9] questioned the IGB relevance due to the relatively small control-subtracted percentage of total body weight loss (%TBWL) and the potential for serious complications. However, more studies will be needed to determine factors influencing the results of IGB and consequently, to adopt a more selective approach for patients who are expected to benefit from IGB insertion.

The aim of this study was to determine the factors affecting weight reduction after intragastric balloon insertion.

## 2. Materials and Methods

This retrospective study was conducted in a specialized hospital for bariatric surgery. It included all the obese individuals who underwent an IGB insertion between January 2016 and June 2019. Patients with the following characteristics were included: aged ≥18 years, with a BMI of more than 25 kg/m^2^ and eligible for intragastric balloon insertion [16], who had failed supervised weight reduction programs (including diet and exercise).

Exclusion criteria included patients younger than 18 years, patients with hiatal hernia, patients with previous gastric surgery, and patients who were not followed up for at least 6 months after removal of the IGB. Patients who were receiving GLP1-antagonists were excluded from our study too.

The hospital records of the patients fulfilling the inclusion criteria were retrieved and the following data were recorded: demographic data, initial BMI, smoking status, co-morbidities, parity [nullipara, para for 1–4 parities and grandpara for more than 4 parities] [17], results of pre-IGB insertion laboratory investigations, complications encountered during the period from insertion to IGB removal, adherence to diet and exercise programs, percentage of excess weight reduction, and BMI at the time of IGB removal.

### 2.1. Outcomes

#### 2.1.1. Primary Endpoints

Correlations between demographic data, initial BMI, compliance to the diet program, adherence to the exercise program, and excess weight loss (EWL) until the time of balloon removal.

#### 2.1.2. Secondary Endpoints

Complications related to IGB insertion.

Regarding handling of missing data after IGB removal, “Listwise Deletion” (complete-case analysis), with removal of all the participants’ data was applied if he/she has one or more missing follow-up values. This technique is commonly used when the researchers are conducting a treatment study and wish to compare the completers analysis.


**Statistical Analysis**


The statistical analysis of data was performed using the Statistical Package for Social Sciences (IBM, SPSS version 25; SPSS Inc., Chicago, IL, USA). Descriptive statistics were applied (i.e., frequency and percentage for categorical variables, in addition to range, mean, and standard deviation for quantitative variables).

The mean percent excess weight loss was calculated for all the patients at the time of IGB extraction with the following formula:Excess weight loss = (no. of kg weight loss) × 100/(excess body weight).
Total body weight loss (TBWL) = pre-op weight − post-op body weight.
% TBWL is the fraction of body weight expressed as a percentage term. (Starting weight minus current weight)/(starting weight) × 100. 

Independent variable *t*-test and F-test were applied to test the significance of observed differences. Moreover, Pearson’s correlation coefficient was calculated between EWL and some quantitative variables. A statistically significant difference was considered at *p* values lower than 0.05.

## 3. Results

The study included 126 patients (108 women and 18 men) who underwent IGB insertion during the period from January 2016 to June 2019 in a specialized hospital for bariatric surgery. The mean age of the patients was 31.7 ± 8.1 years, their mean initial BMI was 35.4 ± 4.8 kg/m^2^, and their initial body weight was 89.83 ± 16.11 kg (range: 67.6–160.0 kg). Regarding the patients’ comorbidities, 3 patients out of 22 (13.6%) who had comorbidities at the time of insertion showed improvement at the time of IGB removal. The demographic and personal characteristics of the patients are listed in Table 1.

Insertion of the balloon in all the patients was performed by an experienced upper gastrointestinal and bariatric surgeon. The ORBERA^®^ Intragastric Balloon System (ORBERA^®^—Apollo Endosurgery, Inc., Austin, TX, USA) was used. The balloon was extracted after a mean duration of 6.9 ± 3.0 months. The mean percentage of EWL at the time of balloon extraction was 55.8 ± 35.7%. The mean TBWL = 12.9 ± 7.6 kg, while the mean TBWL% = 14.2 ± 8.2%. 

Complications occurred in eight patients (6.3%). In two patients (1.59%), the balloons ruptured after 2 and 6 months of insertion, and had to be subsequently extracted. The remaining six patients (4.76%) suffered from gastritis, which was severe in two patients, resulting in balloon extraction after 1 and 2 months, respectively; and mild gastritis in four patients, which was successfully resolved with medical treatment. Regarding the patients’ compliance to the modification program of diet and exercise, there was no significant difference between the young and old patients. Data related to the IG balloon post-insertion and patients’ compliance are shown in Table 2.

The correlation between EWL and patients’ demographic and personal characteristics revealed that patients’ age positively and significantly correlated with their excess weight loss (r = 0.579, *p* = 0.049). In the current study, EWL (as a percentage) showed a significantly negative correlation with both initial weight and initial body mass index of the patients, although there was a significantly greater weight loss (in kilograms) in patients with higher initial BMI values (*p* = 0.008). In addition, there was a significant negative correlation between EWL and the number of pregnancies in female patients (r = −0.238, *p* = 0.013). 

The correlation between EWL and patients’ clinical data revealed a significantly positive correlation between EWL and the duration of balloon insertion (r = 0.332, *p* < 0.001). In addition, there was a significantly lower incidence of EWL among patients who experienced complications. The correlations between EWL and patients’ demographic and clinical data are shown in Table 3 and Table 4.

Multivariate analysis and linear regression proved that the EWL increase significantly with an increase in age and the duration of balloon insertion, and decrease significantly with the initial BMI and an increased number of pregnancies. Linear regression model for excess weight loss is shown in Table 5.

Regarding follow up, after exclusion of the 4 patients who had their IGB removed, 122 patients were followed up. The follow-up period ranged from 7 to 31 months with a mean of 17.5 ± 6.0 months. Out of the 122 patients, 46 patients (37.7%) showed a weight regain 6 months after IGB removal. The mean weight regain after six months: 3.9 ± 2.2 kg. The percent weight changes 6 months after balloon removal (final weight 6 month × 100/weight at balloon removal) = 5.09 ± 2.55% (Figure 1). During the follow-up period, 35 patients (28.7%) underwent bariatric surgeries after balloon removal.

## 4. Discussion

IGBs have been used in the management of obesity since the 1980s.The first generation of IGBs was introduced in 1985 as the Garren-Edwards gastric bubble (GEGB). Nevertheless, the use of the procedure was abandoned due to its significant adverse effects [18,19]. Recent innovations in balloon materials and methods of delivery and extraction, combined with a growing clinical demand, have resulted in IGBs regaining popularity as a weight-loss treatment [20].

In the current study, the ORBERA^®^ Intragastric Balloon System was used in all patients. It was approved by the FDA for use in the USA in August 2015. It is a saline-filled single-balloon system with a fill volume of 500–750 mL. The balloon is placed endoscopically for up to 12 months, and requires endoscopy for deflation and removal [21]. IGB systems differ in their safety, efficacy, and associated adverse effects [16]. According to data from a consensus meeting of Brazilian endoscopists held in Sao Paulo, Brazil, in June 2016, the ORBERA^®^ Intragastric Balloon System is the most widely used balloon worldwide, with over 41,000 reported procedures. The system is associated with a mean total body weight loss of 18.4% ± 2.9% and an adverse event rate of 2.5% after the initial adaptation period. Moreover, it is associated with a 2.2% early-removal rate related to balloon intolerance and with a mortality rate of 0.03% [22]. 

In our study, the mean age of the patients was 31.7 ± 8.1 years, and 85.7% of all the patients were female. Most of the studies reported in the literature have a comparable age and gender distribution to our study. Abeid et al. [23] reported that female patients represented 77% of the studied patients, while the mean age of the studied group was 34.1 ± 10.35 years. In the study by Elia et al. [24], the patients’ mean age was 38 (18–68) years and 75.3% of the patients were women.

The excess weight loss was found to increase significantly with the increase in patient age in our study. As opposed to our findings, Palmisano et al. [25] found no statistically significant correlation between the age and EWL after a 6-month period in a study on 81 patients with a mean age of 45.1 years. In the study by Palmisano et al., two types of devices were used: Heliosphere^®^ new tech (Vienne, France) BAG inflated with air and the BIB™ system, which, together with different endpoints, may explain the difference in the findings between their study and the current study.

In the current study, the initial BMI was 35.4 ± 4.8 kg/m^2^, while the mean initial body weight was 89.83 ± 16.11 kg. We found that the EWL was lower among patients with a higher initial BMI. Palmisano et al. [25] found that the percentage of EWL at the time of IGB removal was significantly higher in patients in the lower BMI categories. Similar results were reported by many other authors [26,27]. On the other hand, some other studies presented outcomes that were different from those of the current study. Alsabah et al. [28] and Ribeiro et al. [29] reported that the lowest EWL in the studied patients was observed among those who had an initial BMI of 35 kg/m^2^, compared to patients with an initial BMI below 30 or above 35 kg/m^2^. 

In their systematic review of 26 primary studies, Yorke et al. [15] reported that the mean EWL decreases as the BMI increases, which is in agreement with our findings. However, the authors found that the mean weight loss in kg is higher in patients in higher BMI categories and concluded that IGB is the most effective in the most obese patients. Although the findings of Yorke et al. [15] resemble our findings regarding the weight loss, we do not agree with them as we believe that it is difficult to draw conclusions based on the weight loss alone as it may produce a false impression of effectiveness. We believe that BMI is the most reliable factor in determining obesity and eligibility of the patient for treatment. Consequently, EWL based on the BMI stems as the best variable for the treatment efficacy evaluation.

In the present study, the mean IGB removal time was at 6.9 ± 3.0 months. The mean EWL was 55.8 ± 35.7% and increased significantly with the increasing duration of the IGB treatment. 

Several studies have shown a mean weight loss associated with ORBERA^®^ IGB therapy comparable to our findings [16,30,31]. Fuller et al. [32] reported that after 3 months, BMI loss was 3.5 kg/m^2^ and the EWL was 36.4%. After 6 months, BMI loss increased to 5.1 kg/m2 and the EWL increased to 50.3%. Palmisano et al. [25] found a strong linear correlation between a 3-month EWL and 6-month results and concluded that the weight reached in the third month appeared to be predictive of the effectiveness of the endoscopic treatment. We believe that the significant proportional correlation between the EWL and duration of IGB treatment seems logical. It is expected that future inventions enabling longer IGB treatments without complications may result in higher efficacy of the treatment.

In this study, complications occurred in eight patients (6.3%). However, only four patients (3.2%) needed early removal of the balloon (two cases required early removal due to balloon rupture and two cases due to severe gastritis), which is comparable to the incidence of complications reported in the literature. Neto et al. [22] reported that the ORBERA^®^ balloon was associated with an adverse event rate of 2.5% after the initial adaptation period. The authors reported a 2.2% early-removal rate due to device intolerance. Abeidet al. [23] reported that 6.4% of the patients suffered from gastritis and 3.6% experienced reflux esophagitis, while 6 patients did not respond to proton pump inhibitors with food intolerance and had their balloons removed. They also reported that one participating patient was diagnosed with pancreatitis and ten patients were diagnosed with gall bladder stones. In the same study, spontaneous deflations occurred in 20 patients. Yorke et al. [15] reported that 3.5% of patients underwent early IGB removal, most commonly due to abdominal pain, nausea/vomiting, balloon deflation, and balloon intolerance (12.0%). They also reported that serious complications were rare, and included gastric ulcers (0.3%), gastric perforations (0.1%), and balloon migration (0.09%).

No mortality was reported in our study. However, Neto et al. [22] and Yorke et al. [15] reported mortality rates of 0.03% and 0.05%, respectively.

In this study, EWL among female patients was found to decrease with higher parity (number of pregnancies). No similar findings have been reported in the literature. Of note, pregnancy has been implicated in future metabolic alterations. Large-scale epidemiological studies have reported associations between increasing parity and the risk of metabolic syndrome [33,34,35,36]. We agree with Saade [37] that a proper understanding of the relationship between changes in maternal metabolism before and after the pregnancy is essential. These changes may provide clues to future health and long-term chronic disease risks.

The limitation of our study relates to a fact that it is a single-center study.

## 5. Conclusions

IGB therapy is a safe and effective option for obesity management, associated with low rates of complications. The EWL after IGB insertion is significantly higher among older patients, those with relatively low initial body mass index, those with a longer duration of IGB insertion, and female patients with less parity. Larger prospective studies are needed to support our results. 

## Figures and Tables

**Figure 1 healthcare-11-00600-f001:**
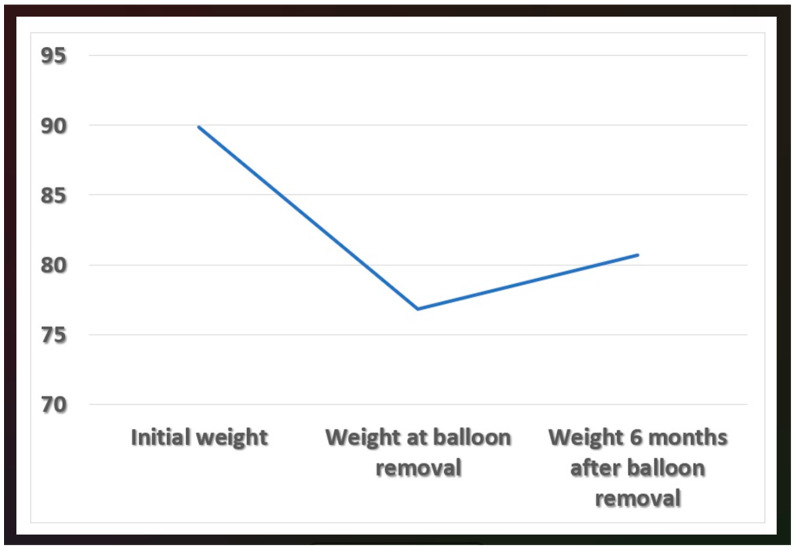
A graph showing the mean initial weight, weight at balloon removal, and 6 months after IGB removal.

**Table 1 healthcare-11-00600-t001:** Demographic and personal characteristics of the studied group.

	No.	%
Age		
Mean ± SD	31.7 ± 8.1 years	
Initial body mass index		
<30 kg/m^2^	8	6.3
30–34.9 kg/m^2^	61	48.4
35–39.9 kg/m^2^	40	31.7
40+ kg/m^2^	17	13.5
Mean ± SD	35.4 ± 4.8 kg/m^2^	
Gender		
Male	18	14.3
Female	108	85.7
Marital status		
Single	61	48.4
Married	59	46.8
Divorced	6	4.8
Parity (*n* = 108)		
None	75	69.4
1–4	25	23.1
>4	8	7.4
Smoking		
Non-smoker	121	96.0
Smoker	5	4.0
Comorbidity		
No	104	82.5
Yes	22	17.5
Past history of abdominal operations		
No	102	81.0
Yes	24	19.0

**Table 2 healthcare-11-00600-t002:** Data related to IGB balloon and post-insertion patients’ compliance.

	No.	%
Duration		
<6 months	33	28.2
6–9 months	71	60.7
>9 months	13	11.1
Mean ± SD	6.9 ± 3.0 months	
Excess weight loss		
<50%	57	45.2
50–75%	36	28.6
>75%	33	26.2
Mean ± SD	55.8 ± 35.7%	
Complications		
No	118	93.7
Yes	8	6.3
Compliance to modification program of diet		
No	76	65.0
Yes	41	35.0
Compliance to modification program of exercise		
No	80	68.4
Yes	37	31.6

**Table 3 healthcare-11-00600-t003:** Participants’ percent excess weight loss (mean ± SD) according to their personal characteristics and clinical data.

Personal Characteristics	Mean of EWL	SD	*p*-Value
Gender			
Male	44.5	30.2	
Female	57.7	36.4	0.147
Marital status			
Single	54.6	29.2	
Married	57.8	42.3	
Divorced	49.2	28.9	0.798
Smoking status			
Smoker	49.3	33.9	
Nonsmoker	56.1	35.9	0.681
Comorbidity			
No	57.3	36.8	
Yes	48.6	29.7	0.297
Past history of abdominal operations			
Yes	52.1	39.3	
No	56.7	35.0	0.575
Initial body mass index (kg/m^2^)			
<30	103.0	68.8	
30–34.9	59.2	32.2	
35–39.9	51.1	26.7	
40+	32.6	22.4	**<0.001 ***
Parity (*n* = 108)			
None (*n* = 75)	60.5	30.5	
1–4 (*n* = 25)	64.7	44.8	
>4 (*n* = 8)	9.1	22.4	**<0.001 ***
Duration			
<6 months	42.3	31.3	
6–9 months	56.4	26.4	
>9 months	88.6	61.6	**<0.001 ***
Complications			
Yes	26.9	27.4	
No	57.8	35.5	**0.018 ***
Compliance modification program of diet			
No	53.9	38.2	
Yes	60.0	29.6	0.376
Compliance modification program of exercise			
No	53.7	34.8	
Yes	61.1	36.6	0.294

* *p* is significant < 0.05.

**Table 4 healthcare-11-00600-t004:** Correlation of different variables with excess weight loss.

Variables	r	*p*-Value
Age	0.579	**0.049 ***
Initial body mass index	−0.413	**<0.001 ***
Initial weight	−0.323	**<0.001 ***
Duration	0.332	**<0.001 ***
Parity (*n* = 108)	−0.238	**0.013 ***

* *p* is significant < 0.05.

**Table 5 healthcare-11-00600-t005:** Linear regression model for excess weight loss.

Variables	B	Standard Error	Standardized Coefficients Beta	*t*-Value	*p*-Value	95% CI for B
Age	1.34	0.41	0.29	3.30	**0.001 ***	(0.53–2.14)
No. of pregnancies	−5.84	1.66	−0.31	−3.53	**0.001 ***	(−9.12–−2.56)
Initial weight	0.80	0.49	0.27	1.64	0.105	(−0.17–1.77)
Initial BMI	−5.06	1.37	−0.61	−3.68	**0.000 ***	(−7.79–−2.34)
Duration of balloon insertion	3.16	0.97	0.28	3.26	**0.001 ***	(1.24–5.08)
Constant	105.98	28.85	--	3.67	**0.000 ***	(48.76–163.20)

* *p* is significant < 0.05.

## Data Availability

Data will be available with the corresponding author to be released on reasonable request.

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
