# Peer review of "Factors Affecting Weight Reduction after Intragastric Balloon Insertion: A Retrospective Study"

_healthcare, 2023, doi:10.3390/healthcare11040600_

Round 1

Reviewer 1 Report

Nice paper detailed experience in IGB. 

Would need to present TBWL and %TBWL too as this is standard practice for most obesity/bariatric papers.

Also, the follow up duration and data is not presented - another key information that is required.

With regards to the poorer outcomes with regards to parity (>4), what would be the reason for this? as there are only a few patients with parity >4, could the authors look into this group and elaborate more please?

Other information worth mentioning would be how many of the patients received medications (GLP1-anatagonists, etc), or even metabolic surgery eventually, as well as remission or improvement in patient co-morbidities.

the tables seem to be copied and pasted from the statistical software and i would advice for the authors to re-format it.

Author Response

Dear Chief Editor

          We would like to thank you and the editorial board for the tremendous efforts you are making in revising and processing our manuscript.

          We also appreciate the valuable comments of the reviewers that raised important points which will improve the quality of the manuscript. The following are our point-by-point responses to the reviewers’ comments:

Reviewer 1:

Would need to present TBWL and %TBWL too as this is standard practice for most obesity/bariatric papers.

  • The TBWL and % TBWL formulas were added to the Methodology section, 8th
  • Mean TBWL (pre-op wt – post-op body wt) = 12.9± 7.6 kg, while the Mean TBWL% (Starting wt - current wt)/(starting wt)x100 = 14.2±8.2%. This was added to the results section, 2nd paragraph, 5th

Also, the follow-up duration and data are not presented - another key information that is required.

  • In our study, the minimal follow-up duration was 6 months and this was clarified in the exclusion criteria (Materials and Methods section, 2nd paragraph, 2nd line)
  • Duration of follow-up was added to the Results section, 6th Important follow-up data like weight regain and the percentage of patients who underwent bariatric surgery after IGB removal was added as per the reviewer’s suggestion.

With regards to the poorer outcomes with regards to parity (>4), what would be the reason for this? as there are only a few patients with parity >4, could the authors look into this group and elaborate more, please?

  • This finding was observational in our study, despite the small number of patients (8 out of 126 patients). We went deep through the literature but we could not find similar findings.
  • We discussed this issue in the last paragraph of the discussion section (11th paragraph), we could find few reports describing the relationship between metabolic changes and pregnancy, however, the small number of patients who had >4 pregnancies prevented us from making a strong conclusion, so, we recommended in our conclusion future studies with a larger number of patients to explore more about this finding.

Other information worth mentioning would be how many patients received medications (GLP1-antagonists, etc), or even metabolic surgery eventually, as well as remission or improvement in patient co-morbidities.

  • Regarding the GLP1-antagonists, we thank the reviewer for his valuable comments. None of our patients received GLP1-antagonist medications before balloon insertion as it was not included in the insurance during the duration the data was collected. Also, we thought they might confuse our results. This was added to the exclusion criteria to be clear to the reader (Materials and Methods section, 2nd paragraph, 3rd line).
  • The follow-up of our patients according to their medical files was added to the results section, 6th The follow-up data included weight regain, and % of patients who underwent metabolic surgeries.

the tables seem to be copied and pasted from the statistical software and I would advise the authors to re-format them.

  • Table number 5 was reformatted as per the reviewer’s suggestion. If the reviewer still has any comments, please specify.

Table 5 Linear regression model for excess weight loss              

Variables

B

Standard Error

Standardized coefficients

Beta

t-value

P

value

95% CI for B

Age

1.34

0.41

0.29

3.30

0.001*

(0.53 - 2.14)

No. of pregnancies

-5.84

1.66

-0.31

-3.53

0.001*

(-9.12 - -2.56)

Initial weight   

0.80

0.49

0.27

1.64

0.105

(-0.17 - 1.77)

Initial BMI      

-5.06

1.37

-0.61

-3.68

0.000*

(-7.79 - -2.34)

Duration of balloon insertion

3.16

0.97

0.28

3.26

0.001*

(1.24 - 5.08)

Constant

105.98

28.85

--

3.67

0.000*

(48.76 - 163.20)

*P is significant < 0.05

Reviewer 2 Report

The paper is clear, and well written, easy to read.

However the content of the parer adds little to existing litterature. The patient inclusion is some time ago and therefore a longer term follow-up with regard to weightloss would perhaps increase its scientific content. 

1. What is the main question addressed by the research?

        Short term utcome after gastric balloon

2. Do you consider the topic original or relevant in the field? Does it address a specific gap in the field?

        Non-orginal topic, adds little to pervio results. No clear gap is addressed

3. What does it add to the subject area compared with other published material?

        No new data

4. What specific improvements should the authors consider regarding the methodology? What further controls should be considered?

        Long term outcomes with regard to weight loss

5. Are the conclusions consistent with the evidence and arguments presented and do they address the main question posed?

        yes

6. Are the references appropriate?

        Yes but could be updated

7. Please include any additional comments on the tables and figures.

        Fine, nothing to add.

Author Response

Dear Chief Editor

          We would like to thank you and the editorial board for the tremendous efforts you are making in revising and processing our manuscript.

          We also appreciate the valuable comments of the reviewers that raised important points which will definitely improve the quality of the manuscript. The following are our point-by-point responses to the reviewers’ comments:

Reviewer 2:

However, the content of the paper adds little to the existing literature. The patient inclusion is some time ago and therefore a longer-term follow-up with regard to weight loss would perhaps increase its scientific content. 

  1. What is the main question addressed by the research?

        Short term outcome after gastric balloon

  • The aim of our research was to determine the factors affecting weight reduction after intragastric balloon insertion as the primary endpoint. The recommended duration of insertion of IGB at the time of the research was 6 months. We had some patients who had their IGB for a longer duration. So we disagree with the reviewer regarding describing our study as a short-term outcome.
  • We agree with the reviewer that the data related to the follow-up period were unclear. The follow-up of our patients according to their medical files was added to the results section, 6th The follow-up data included weight regain, and % of patients who underwent metabolic surgeries.

  1. Do you consider the topic original or relevant in the field? Does it address a specific gap in the field?

        Non-original topic, adds little to previous results. No clear gap is addressed

  • Our study achieved the aim by determining the factors affecting weight reduction after IGB insertion. It highlighted the relationship between parity and weight reduction which was not reported before and still needs further studies to figure out.

  1. What does it add to the subject area compared with other published material?

        No new data

  • Our study achieved the aim by determining the factors affecting weight reduction after IGB insertion. It highlighted the relationship between parity and weight reduction which was not reported before and still needs further studies to figure out.

  1. What specific improvements should the authors consider regarding the methodology? What further controls should be considered?

        Long-term outcomes with regard to weight loss

  • We agree with the reviewer that the data related to the follow-up period were unclear. The follow-up of our patients according to their medical files was added to the results section, 6th The follow-up data included weight regain, and % of patients who underwent metabolic surgeries.
  • Nevertheless, being a retrospective study is one limitation of long-term follow-up. In addition, long-term follow-up was not the primary endpoint of our study.

  1. Are the conclusions consistent with the evidence and arguments presented and do they address the main question posed?

  Yes

  • We thank the reviewer for his comment.

  1. Are the references appropriate?

      Yes but could be updated

  • We revised the list of references, one old reference was removed as per the reviewer's suggestion and the numbering was rearranged accordingly.

  1. Please include any additional comments on the tables and figures.

  Fine, nothing to add.

  • We thank the reviewer for his comment.

Round 2

Reviewer 1 Report

thank you for the edits.

Author Response

Dear Chief Editor

          We want to express our gratitude to you and the entire editorial board for all of your hard work editings and processing our article. We also value the reviewers' insightful remarks, which brought up some crucial issues and undoubtedly increased the quality of the article.

Our detailed responses to the reviewers' remarks are as follows:

Reviewer 1 (Round 2):

Thank you for the edits.

  • Thank you for your previous valuable remarks

Reviewer 2 Report

Thank you for the opportunity to re-review this paper. I believe many of the comments have been addressed, In the written response. 

Comments:

The results need to be more clearly divided into the primary endpoint (Factors affecting weightloss during the time until IGB removal). Then a clear division into the "long term" follow-up results. 

Regarding long-term follow-up: " The mean weight regains after six months: 3.9±2.2 kg. During the follow-up period,35 patients (28.7%) underwent bariatric surgeries after balloon removal" this value of kg is irrelevant; relevant would be a comparison to weight loss during IBD. please update. 

- Long-term follow-up regarding weight regain could be presented visually. 

How missing data was handled is not described, and this needs to be addressed.  This is important. 

Methods,

Missing header "statistics"

Results:

- Complications in its own paragraph

- Comorbidities move to the patient demographic section

- a significant relationship with the patient's age (r=0.579, P=0.049), bold statement with a P value of 0.049 - refrase. 

Table 3.

Complications • Yes 26.9 27.4 • No 57.8 35.5 0.018* - What is presented here?

Author Response

Dear Chief Editor

          We want to express our gratitude to you and the entire editorial board for all of your hard work editings and processing our article. We also value the reviewers' insightful remarks, which brought up some crucial issues and undoubtedly increased the quality of the article.

Our detailed responses to the reviewers' remarks are as follows:

Reviewer (Round 2):

Thank you for the opportunity to re-review this paper. I believe many of the comments have been addressed, In the written response. 

Comments:

The results need to be more clearly divided into the primary endpoint (Factors affecting weight loss during the time until IGB removal). Then a clear division into the "long term" follow-up results. 

  • The paragraphs of the results are arranged as follows:
    • 1st: Demographic data and Comorbidities (comorbidities were added as per the reviewer's suggestion).
    • 2nd: Data regarding balloon insertion, weight loss, and extraction.
    • 3rd: Complications (was separated to be in a separate paragraph as per the reviewer’s suggestion)
    • 4th: Correlations between the EWL and patients’ demographic and personal characteristics.
    • 5th: Correlations between the EWL and patients’ clinical data.
    • 6th: Results of the multivariate analysis.
    • 7th: Follow-up ( The paragraph starts with “Regarding follow-up” to make it clear as per the reviewer’s suggestion).
  • We think this arrangement is reasonable for presenting our results. If the reviewer has any comments, please specify.

Regarding long-term follow-up: " The mean weight regains after six months: 3.9±2.2 kg. During the follow-up period,35 patients (28.7%) underwent bariatric surgeries after balloon removal" this value of kg is irrelevant; relevant would be a comparison to weight loss during IBD. please update. 

  • The percent weight change 6 months after balloon removal (final weight 6 months x 100 / weight at balloon removal) = 5.09±2.55%. This was added to the Results section, 7th paragraph, line 5.

- Long-term follow-up regarding weight regain could be presented visually. 

  • We think this is a very nice remark, thank you. A graph was added to make the weight changes before, during, and after the removal of the balloon clear to the reader.

How missing data was handled is not described, and this needs to be addressed.  This is important. 

  • Regarding the handling of missing data after IGB removal, “Listwise Deletion” (complete-case analysis), with the removal of all participant’s data was applied if he/she has one or more follow-up missing values.  This technique is commonly used when researchers are conducting a treatment study and wish to compare the completers' analysis. This was added to the Methodology section, 4th paragraph as per the reviewer’s suggestion.

Methods,

Missing header "statistics"

  • The header was added as per the reviewer’s suggestion.

Results:

- Complications in its own paragraph

  • Complications were separated to be in their own paragraph as per the reviewer’s suggestion.

- Comorbidities move to the patient demographic section

  • The comorbidities section was added to the demographic section as suggested by the reviewer (Results section, 1st paragraph, line 4).

- a significant relationship with the patient's age (r=0.579, P=0.049), bold statement with a P value of 0.049 - rephrase. 

  • The sentence was rephrased to be “Patients’ age positively and significantly correlated with their excess weight loss (r=0.579, p=0.049)”. This was added to the Results section, 4th paragraph, 2nd

Table 3.

Complications • Yes 26.9 27.4 • No 57.8 35.5 0.018* - What is presented here?

  • Percent excess weight loss (Mean±SD) for patients who had complications (26.9±27.4 kg) compared with those who had no complications (57.8±35.5), with a significant p-value = 0.018